# On Gap-Based Lower Bounding Techniques for Best-Arm Identification

**DOI:** 10.3390/e22070788

**Published:** 2020-07-20

**Authors:** Lan V. Truong, Jonathan Scarlett

**Affiliations:** 1Department of Engineering, University of Cambridge, Cambridge CB2 1PZ, UK; 2Department of Computer Science & Department of Mathematics, National University of Singapore, Singapore 117418, Singapore; scarlett@comp.nus.edu.sg

**Keywords:** multi-armed bandits, best-arm identification, information-theoretic lower bounds, PAC learning

## Abstract

In this paper, we consider techniques for establishing lower bounds on the number of arm pulls for best-arm identification in the multi-armed bandit problem. While a recent divergence-based approach was shown to provide improvements over an older gap-based approach, we show that the latter can be refined to match the former (up to constant factors) in many cases of interest under Bernoulli rewards, including the case that the rewards are bounded away from zero and one. Together with existing upper bounds, this indicates that the divergence-based and gap-based approaches are both effective for establishing sample complexity lower bounds for best-arm identification.

## 1. Introduction

The multi-armed bandit (MAB) problem [1] provides a versatile framework for sequentially searching for high-reward actions, with applications including clinical trials [2], online advertising [3], adaptive routing [4], and portfolio design [5]. The best-arm identification problem seeks to find the arm with the highest mean using as few arm pulls as possible, and dates back to the works of Bechhofer [6] and Paulson [7]. More recently, several algorithms have been proposed for best-arm identification, including successive elimination [8], lower-upper confidence bound algorithms [9,10], PRISM [11], and gap-based elimination [12]. The latter establishes a sample complexity that is known to be optimal in the two-arm case [13], and more generally near-optimal.

Complementary to these upper bounds is information-theoretic lower bounds on the performance of any algorithm. Such bounds serve as a means to assess the degree of optimality of practical algorithms, and identify where further improvements are possible, thus focusing research towards directions that can have the greatest practical impact. Lower bounds were given by Mannor and Tsitsiklis [14] for Bernoulli bandits, and by Kaufmann et al. [15] for more general reward distributions. Both of these works were based on the difficulty of distinguishing bandit instances that differ in only a single arm distribution, but the subsequent analysis techniques differed significantly, with [14] using a direct change-of-measure analysis and introducing gap-based quantities equaling the difference between two arm means, and [15] using a form of the data processing inequality for KL divergence. We refer to these as the gap-based and divergence-based approaches, respectively. Further works on best-arm identification lower bounds include [16,17,18].

The divergence-based approach was shown in [15] to attain a stronger result than that of [14] with a simpler proof, as we outline in Section 2.2. In this paper, we address the question of whether the gap-based approach is fundamentally limited, or can be refined to attain a similar results to [15]. We show that the correct answer is the latter in many cases of interest, by suitable refinements of the analysis of [14]. The existing results and our results are presented in Section 2, and our analysis is presented in Section 3.

## 2. Overview of Results

### 2.1. Problem Setup

We consider the following setup:There are *M* arms with Bernoulli rewards; the means are $p=(p_{1},p_{2},\cdots ,p_{M})$, and this set of means is said to define the bandit instance. Our analysis will consider instances with arms sorted such that $p_{1}\geq p_{2}\cdots \geq p_{M}$, without loss of generality.The agent would like to find an arm whose arm mean is within ϵ of the highest arm mean for some 0<ε<1, i.e., $p_{l}>p_{1}-\varepsilon$. Even if there are multiple such arms, just identifying one of them is good enough.In each round, the agent can pull any arm l∈[M] and observe an reward $X_{l}^{\left(s\right)}∼Bernoulli\left(p_{l}\right)$, where *s* is the number of times the *l*-th arm has been pulled so far. We assume that the rewards are independent, both across arms and across times.In each round, the agent can alternatively choose to terminate and output an arm index $\hat{l}$ believed to be ϵ-optimal. The index at which this occurs is denoted by *T*, and is a random variable because it is allowed to depend on the rewards observed. We are interested in the expected number of arm pulls (also called the sample complexity) $E_{p}\left[T\right]$ for a given instance p, which should ideally be as low as possible.An algorithm is said to be (ε,δ)-PAC (Probably Approximately Correct) if, for all bandit instances, it outputs an ε-optimal arm with probability at least 1−δ when it terminates at the stopping time *T*.

We will frequently make use of some fundamental quantities. First, the best arm mean and the gap to the best arm are denoted by
(1)$$\begin{array}{cc} p_{*} & :=p_{1}, \end{array}$$
(2)$$\begin{array}{cc} \Delta_{l} & :=p_{*}-p_{l}. \end{array}$$
The set of ϵ-optimal arms and the set of ϵ-suboptimal arms are respectively given by
(3)$$M(p,\varepsilon ):=\{l\in \left[M\right]:p_{l}>p_{*}-\varepsilon \},$$
(4)$$N(p,\varepsilon ):=\{l\in \left[M\right]:p_{l}\leq p_{*}-\varepsilon \},$$
and we make use of the binary KL divergence function
(5)$$\mathrm{KL}\left(p,q\right):=p\log\frac{p}{q}+\left(1-p\right)\log\frac{1-p}{1-q},$$
where here and subsequently, log(·) denotes the natural logarithm.

### 2.2. Existing Lower Bounds

For any fixed $\underline{p}\in \left(0,1/2\right)$, Mannor and Tsitsiklis [14] showed that if an algorithm is (ϵ,δ)-PAC with respect to all instances with $\min_{l}p_{l}\geq \underline{p}>0$, and if $\epsilon \leq \frac{1-p^{*}}{4}$ and $\delta \leq e^{-8}/8$, then for any constant α∈(0,2), there exists $c_{1}=O\left({\underline{p}}^{2}\right)$ (depending on α) such that
(6)$$\begin{array}{c} E_{p}\left[T\right]\geq c_{1}\left[\frac{{\left(|\tilde{M}\left(p,\varepsilon \right)|-1\right)}^{+}}{\varepsilon^{2}}+\sum_{l\in \tilde{N}\left(p,\varepsilon \right)}\frac{1}{\Delta_{l}^{2}}\right]\log\frac{1}{8\delta } \end{array}$$
where
(7)$$\begin{array}{c} \tilde{M}\left(p,\varepsilon \right)=M\left(p,\varepsilon \right)\cap \left\{l\in \left[M\right]:p_{l}\geq \frac{\varepsilon +p_{*}}{2-\alpha }\right\}, \end{array}$$
(8)$$\begin{array}{c} \tilde{N}\left(p,\varepsilon \right)=N\left(p,\varepsilon \right)\cap \left\{l\in \left[M\right]:p_{l}\geq \frac{\varepsilon +p_{*}}{2-\alpha }\right\}. \end{array}$$

Note that the subsets $\tilde{M}\left(p,\varepsilon \right)$ and $\tilde{N}\left(p,\varepsilon \right)$ do not always form a partition of the arms, i.e., it may hold that $\tilde{M}\left(p,\varepsilon \right)\cup \tilde{N}\left(p,\varepsilon \right)⊊\left[M\right]$. The sets increase in size as α decreases, but implicitly this leads to a lower value of $c_{1}$. In addition, as we will see below, the ${\underline{p}}^{2}$ dependence entering via $c_{1}$ is not necessary.

We also note that the lower bound in (Equation 6) depends on the instance-specific quantities $\tilde{M}\left(p,\varepsilon \right)$, $\tilde{N}\left(p,\varepsilon \right)$, and $\Delta_{l}$, and is thus an instance-dependent bound. On the other hand, the lower bound is only stated for (ε,δ)-PAC algorithms, and the PAC guarantee requires the algorithm to eventually succeed on any instance (subject to the assumptions given on $p_{l}$, ϵ, and δ).

Kaufmann et al. [15] improved Mannor and Tsitsiklis’s lower bound by using a form of data processing inequality for KL divergence, leading to the following whenever δ≤0.15 and $0<\varepsilon <\min\{p_{*},1-p_{*}\}$ [15] (Remark 5):(9)$$\begin{array}{c} E_{p}\left[T\right]\geq \left[\frac{|M(p,\varepsilon )|-1}{\mathrm{KL}(p_{*}-\varepsilon ,p_{*}+\varepsilon )}+\sum_{l\in N(p,\varepsilon )}\frac{1}{\mathrm{KL}(p_{l},p_{*}+\varepsilon )}\right]\log\frac{1}{2.4\delta }. \end{array}$$
To directly compare this result with (Equation 6), it is useful to apply the following inequality [19] (Equation (2.8)):(10)$$\begin{array}{c} 2{\left(p-q\right)}^{2}\leq \mathrm{KL}\left(p,q\right)\leq \frac{{\left(p-q\right)}^{2}}{q(1-q)}, \end{array}$$
which yields
(11)$$\begin{array}{c} E_{p}\left[T\right]\geq \left(p^{*}+\epsilon \right)\left(1-p^{*}-\epsilon \right)\left[\frac{|M(p,\varepsilon )|-1}{4\epsilon^{2}}+\sum_{l\in N(p,\varepsilon )}\frac{1}{{\left(\varepsilon +\Delta_{l}\right)}^{2}}\right]\log\frac{1}{2.4\delta }. \end{array}$$
Even this weakened bound can significantly improve on (Equation 6), since (i) $M\left(p,\varepsilon \right)⊃\tilde{M}\left(p,\varepsilon \right)$ and $N\left(p,\varepsilon \right)⊃\tilde{N}\left(p,\varepsilon \right)$, (ii) the ${\underline{p}}^{2}$ dependence is replaced by $\left(p^{*}+\epsilon \right)\left(1-p^{*}-\epsilon \right)$, so the dependence on the smallest arm mean is avoided (The $1-p^{*}-\epsilon$ term is potentially small when ϵ is close to $1-p^{*}$, but since (Equation 6) assumes $\epsilon \leq \frac{1-p^{*}}{4}$, we can still say that (Equation 11) is at least as good as (Equation 6)), and (iii) the assumption $\epsilon \leq \frac{1-p^{*}}{4}$ is avoided.

### 2.3. Our Result and Discussion

Our lower bound, stated in the following theorem, is developed based on Mannor and Tsitsiklis’s analysis for best-arm identification [14] (Theorem 1), but uses novel refinements of the techniques therein to further optimize the bound (see Appendix C for an overview of these refinements).

**Theorem** **1.**
*For any bandit instance $p\in {\left(0,p_{*}\right]}^{M}$ with $p_{*}\in \left(0,1\right)$, and any (ε,δ)-PAC algorithm with $0<\varepsilon <1-p_{*}$ and $0<\delta <\delta_{0}$ for some $\delta_{0}<1/4$, we have*
(12)$$\begin{array}{cc} E_{p}\left[T\right] & \geq \frac{2\gamma_{0}\left(p_{*}+\varepsilon \right)\left(1-p_{*}-\varepsilon \right)}{7(\xi +1)}\left[\frac{|M(p,\varepsilon )|-1}{4\varepsilon^{2}}+\sum_{l\in N(p,\varepsilon )}\frac{1}{{\left(\varepsilon +\Delta_{l}\right)}^{2}}\right]\log\frac{1+4\delta_{0}}{4\delta }, \end{array}$$
*where*
(13)$$\begin{array}{cc} \gamma_{0} & =\frac{1-4\delta_{0}}{8}, \end{array}$$
(14)$$\begin{array}{cc} \theta & =\frac{2\delta }{1-4\gamma_{0}}=\frac{4\delta }{1+4\delta_{0}}, \end{array}$$
*and ξ>0 is the unique positive solution of the following quadratic equation:*
(15)$$\begin{array}{c} 7\gamma_{0}\xi^{2}\log\frac{1}{\theta }=3\left(\xi +1\right). \end{array}$$


Observe that this result matches (Equation 11) (with modified constants), and therefore exhibits the above benefit of depending on the full sets M and N without the condition $p_{l}\geq \frac{\varepsilon +p_{*}}{2-\alpha }$ (see (Equation 7)–(8)), as well as avoiding the dependence on $\underline{p}$, and permitting the broadest range of ϵ and δ among the above results.

The result (Equation 11) in turn matches (Equation 9) whenever the right-hand inequality in (Equation 10) is tight (i.e., whenever $\mathrm{KL}\left(p,q\right)=\Theta \left(\frac{{\left(p-q\right)}^{2}}{q(1-q)}\right)$). This is clearly true when *p* and *q* (representing the arm means) are bounded away from zero and one, and also in certain limiting cases approaching these endpoints (e.g., when *p* and *q* both tend to one, but $\frac{1-p}{1-q}=\Theta \left(1\right)$). However, there are also limiting cases where the upper bound in (Equation 10) is not tight (e.g., $p=1-\sqrt{\eta }$ and q=1−η as η→0), and in such cases, the bound (Equation 9) remains tighter than that of Theorem 1.

## 3. Proof of Theorem 1

We follow the general steps of (Theorem 5 [14]), but with several refinements to improve the final bound. The main differences are outlined in Appendix C.


*Step 1: Defining a Hypothesis Test*


Let us denote the true (unknown) expected reward of each arm by $Q_{l}$ for all l∈[M]. Similarly to [14,15], we consider *M* hypotheses as follows:(16)$$\begin{array}{c} H_{1}:Q_{l}=p_{l}, \end{array}\;\begin{array}{c} \forall l\in \left[M\right], \end{array}$$
and for each l≠1,
(17)$$\begin{array}{c} H_{l}:Q_{l}=p_{*}+\varepsilon , \end{array}\;\begin{array}{c} Q_{l^{\prime }}=p_{l^{\prime }} \end{array}\;\begin{array}{c} \forall l^{\prime }\in \left[M\right]\backslash \left\{l\right\}. \end{array}$$
If hypothesis $H_{l}$ is true, the (ϵ,δ)-PAC algorithm must return arm *l* with probability at least 1−δ. We will bound sample complexity when the hypothesis $H_{l}$ is true. We denote by $E_{l}$ and $P_{l}$ the expectation and probability, respectively, under hypothesis $H_{l}$.

Let $B_{l}$ be the event that the algorithm returns arm *l*. Since $\sum_{l\in M(p,\varepsilon )}P_{1}\left(B_{l}\right)\leq 1$ and |M(p,ε)|≥1, there is at most one arm $l_{0}\in M\left(p,\varepsilon \right)$ that satisfies $P_{1}\left(B_{l_{0}}\right)>\frac{1}{2}$. Defining
(18)$$\begin{array}{c} M_{0}\left(p,\varepsilon \right):=\left\{l\in M\left(p,\varepsilon \right):P_{1}\left[B_{l}\right]\leq \frac{1}{2}\right\}=\left\{l\in M\left(p,\varepsilon \right):l\neq l_{0}\right\}, \end{array}$$
it follows that
(19)$$\begin{array}{c} |M_{0}{\left(p,\varepsilon \right)|\geq (|M\left(p,\varepsilon \right)|-1)}^{+}. \end{array}$$

Define
(20)$$\begin{array}{c} T\left(p,\varepsilon \right):=M_{0}\left(p,\varepsilon \right)\cup N\left(p,\varepsilon \right), \end{array}$$
as well as
(21)$$\begin{array}{c} B_{M(p,\varepsilon )}:=\underset{l\in M(p,\varepsilon )}{⋃}B_{l}, \end{array}$$
which is the event that the policy eventually select an arm in the ε-neighborhood of the best arm in [M]. Since the policy is (ε,δ)-correct with $\delta <\delta_{0}$, we must have
(22)$$\begin{array}{c} P_{1}\left[B_{M(p,\varepsilon )}\right]\geq 1-\delta >1-\delta_{0}, \end{array}$$
and it follows from (Equation 18) and (Equation 22) that
(23)$$\begin{array}{cc} P_{1}\left[B_{l}\right]\; & \leq \max\left\{\delta_{0},1/2\right\} \end{array}$$
(24)$$\begin{array}{cc} \; & =\frac{1}{2} \end{array}$$
for all l∈T(p,ε).


*Step 2: Bounding the Number of Pulls of Each Arm*


Before proceeding, we make some additional definitions: (25)$$\begin{array}{cc} \alpha_{l} & :=\frac{\varepsilon +\Delta_{l}}{\left(1-p_{l}\right)p_{l}}, \end{array}$$(26)$$\begin{array}{cc} \beta_{l} & :=\alpha_{l}\sqrt{\frac{1-p_{l}}{1-(p_{*}+\varepsilon )}}, \end{array}$$(27)$$\begin{array}{cc} {\tilde{\alpha }}_{l} & :=\alpha_{l}-\frac{4}{3}{\left(\alpha_{l}\left(1-p_{l}\right)\right)}^{2}, \end{array}$$(28)$$\begin{array}{cc} {\tilde{\beta }}_{l} & :=\beta_{l}-\frac{4}{3}{\left(\alpha_{l}\left(1-p_{l}\right)\right)}^{2}, \end{array}$$
The definitions (27) and (28) will only be used for arms with $\frac{\varepsilon +\Delta_{l}}{p_{l}}\leq \frac{1}{2}$, and for such arms, we will establish in the analysis that ${\tilde{\alpha }}_{l}\geq 0$ and ${\tilde{\beta }}_{l}\geq 0$.

We prove the following lemma, characterizing the probability of a certain event in which (i) the number of pulls of some arm l∈T(p,ε) falls below a suitable threshold (event $A_{l}$ below), (ii) a deviation bound holds regarding the number of observed 1’s from pulling arm *l* (event $C_{l}$ below), and (iii) arm *l* is not returned (event $B_{l}^{c}$).

**Lemma** **1.**
*For each l∈[M], let $T_{l}$ be the total number of times that arm l is pulled under the (ε,δ)-correct policy. Let $K_{l}=X_{l}^{\left(1\right)}+X_{l}^{\left(2\right)}+\cdots +X_{l}^{\left(T_{l}\right)}$ be the total number of unit rewards obtained from pulling the arm l up to the $T_{l}$-th time. Let*
(29)$$\begin{array}{c} G_{1,l}:=\frac{7p_{l}^{2}\alpha_{l}^{2}{\left(1-p_{l}\right)}^{2}}{2\left(p_{*}+\varepsilon \right)\sqrt{\left(1-p_{l}\right)\left(1-\left(p_{*}+\varepsilon \right)\right)}}T_{l}, \end{array}$$
(30)$$\begin{array}{c} G_{2,l}:=\left[{\tilde{\beta }}_{l}\left(p_{l}T_{l}-K_{l}\right)1\left\{p_{l}T_{l}>K_{l}\right\}+{\tilde{\alpha }}_{l}\left(p_{l}T_{l}-K_{l}\right)1\left\{p_{l}T_{l}\leq K_{l}\right\}\right]1\left\{\frac{\varepsilon +\Delta_{l}}{p_{l}}\leq \frac{1}{2}\right\}, \end{array}$$
*where $\alpha_{l}$, ${\tilde{\alpha }}_{l}$, and ${\tilde{\beta }}_{l}$ are defined in *(Equation 25)*, *(27)*, and *(28)*, respectively. Let*
(31)$$\begin{array}{c} \nu_{l}:=\left(\xi +1\right)\left(\sqrt{\frac{1-p_{l}}{1-p_{*}-\varepsilon }}\right), \end{array}$$
*where ξ is defined in *(Equation 15)*. Define the following events:*
(32)$$\begin{array}{c} A_{l}:=\left\{G_{1,l}\leq \frac{1}{\nu_{l}}\log\frac{1}{\theta }\right\}, \end{array}$$
(33)$$\begin{array}{c} C_{l}=\left\{G_{2,l}\leq \frac{\xi }{\nu_{l}}\left(\sqrt{\frac{1-p_{l}}{1-p_{*}-\varepsilon }}\right)\log\frac{1}{\theta }\right\}, \end{array}$$
(34)$$\begin{array}{c} S_{l}:=A_{l}\cap B_{l}^{c}\cap C_{l}. \end{array}$$
*If l∈T(p,ε) (see *(Equation 20)*), then under the condition that*
(35)$$\begin{array}{c} E_{1}\left[G_{1,l}\right]<\frac{\gamma_{0}}{\nu_{l}}\log\frac{1}{\theta }, \end{array}$$
*we have*
(36)$$\begin{array}{c} P_{1}\left[S_{l}\right]>\frac{1-4\gamma_{0}}{2}. \end{array}$$


**Proof.** See Appendix A. □

Intuitively, $A_{l}$ is the event that the total number of times that arm *l* is pulled is small, and $C_{l}$ is the event that $|p_{l}T_{l}-K_{l}|$ is not too large (since pulling an arm $T_{l}$ times should produce roughly $p_{l}T_{l}$ ones). The lemma indicates that if $E[T_{l}]$ is not too large, then $P[A_{l}\cap B_{l}^{c}\cap C_{l}]$ is lower bounded, and this will ultimately lead to a lower bound on $P[B_{l}^{c}]$, the event of primary interest.

In Lemma 2 below, we will use Lemma 1 to deduce a lower bound on $E_{1}\left[G_{1,l}\right]$, which amounts to a lower bound on the average number of arm pulls by the definition of $G_{1,l}$. Before doing so, we introduce a likelihood ratio that will be used in a change-of-measure argument [14].

For any given time t≥1 and l∈[M], let $T_{l}\left(t\right)$ be the total number of times that arm *l* is pulled by time *t*. Define
(37)$$\begin{array}{c} X_{l}^{T_{l}\left(t\right)}:=\left\{X_{l}^{\left(1\right)},X_{l}^{\left(2\right)},\cdots ,X_{l}^{\left(T_{l}\left(t\right)\right)}\right\}, \end{array}$$
and let
(38)$$\begin{array}{c} F_{t}:=\sigma \left(X_{1}^{T_{1}\left(t\right)},X_{2}^{T_{2}\left(t\right)},\cdots ,X_{M}^{T_{M}\left(t\right)}\right) \end{array}$$
be the σ-algebra generated by $X_{1}^{T_{1}\left(t\right)},X_{2}^{T_{2}\left(t\right)},\ldots ,X_{M}^{T_{M}\left(t\right)}$ for all t=1,2,….

Recall that *T* is the stopping time of the algorithm, and that $T_{l}:=T_{l}\left(T\right)$ for all l∈[M]. Moreover, let $W=F_{T}$ be the entire history up to the stopping time *T*. We define the following likelihood ratio:(39)$$\begin{array}{c} L_{l}\left(w\right)=\frac{P_{l}\left(W=w\right)}{P_{1}\left(W=w\right)} \end{array}$$
for every possible history *w*. Moreover, we let $L_{l}\left(W\right)$ denote the corresponding random variable. Given the history up to time T−1 (i.e., $F_{T-1}$), the arm reward at time *T* has the same probability distribution under $H_{1}$ and $H_{l}$ unless the chosen arm is arm *l*. Therefore, we have
(40)$$\begin{array}{c} L_{l}\left(W\right)=\frac{{\left(p_{*}+\varepsilon \right)}^{K_{l}}{\left(1-p_{*}-\varepsilon \right)}^{T_{l}-K_{l}}}{p_{l}^{K_{l}}{\left(1-p_{l}\right)}^{T_{l}-K_{l}}}, \end{array}$$
where $K_{l}:=X_{l}^{\left(1\right)}+X_{l}^{\left(2\right)}+\cdots +X_{l}^{\left(T_{l}\right)}$ (or the total number of 1’s in the $T_{l}$ pulls of the arm *l*).

The following proposition presents one of our key technical results towards establishing the lower bound. We use the definitions in (Equation 1)–(Equation 5), along with (Equation 25)–(28).

**Proposition** **1.**
*Fix the bandit instance p, the parameter $0<\varepsilon <1-p_{*}$, and the history W with corresponding values $K_{l}$ and $T_{l}$. Recalling the definitions of $\alpha_{l}$, ${\tilde{\alpha }}_{l}$, and ${\tilde{\beta }}_{l}$ in *(Equation 25)*, *(27)*, and *(28)*, respectively, we have*
(41)$$\begin{array}{c} L_{l}\left(W\right)\geq \exp\left(-G_{1,l}-G_{2,l}\right), \end{array}$$
*where $G_{1,l}$ and $G_{2,l}$ are defined in *(Equation 29)*–*(30)*.*


**Proof.** See Appendix B. □

Based on Lemma 1 and Proposition 1, we obtain the following extension of [14] (Lemma 6) lower bounding the average of each $G_{1,l}$; this lower bound will later be translated to a lower bound on the number of arm pulls $T_{l}$.

**Lemma** **2.**
*For any arm l∈T(p,ε), the following holds:*
(42)$$\begin{array}{c} E_{1}\left[G_{1,l}\right]\geq \frac{\gamma_{0}}{\nu_{l}}\log\frac{1}{\theta }, \end{array}$$
*where θ and $\nu_{l}$ are defined in *(14)* and *(Equation 31)*, respectively.*


**Proof.** We use a proof by contradiction. Assume that
(43)$$\begin{array}{c} E_{1}\left[G_{1,l}\right]<\frac{\gamma_{0}}{\nu_{l}}\log\frac{1}{\theta }, \end{array}$$
then by Lemma 1, Equation (Equation 36) holds. Moreover, by Proposition 1, we have
(44)$$\begin{array}{c} L_{l}\left(W\right)\geq \exp\left(-G_{1,l}-G_{2,l}\right), \end{array}$$
and recalling the definition of $S_{l}$ in (34), it follows from (Equation 44) that
(45)Ll(W)1Sl≥exp−G1,l−G2,l1Sl(46)≥exp−1νlξ1−pl1−p*−ε+1log1θ1Sl(47)≥exp−1νl(ξ+1)1−pl1−p*−εlog1θ1Sl
where (46) follows from the definitions in (Equation 32)–(33), and (47) follows from the fact that $1-p_{l}\geq 1-p_{*}\geq 1-p_{*}-\varepsilon$ for all l∈[M].By the choice of $\nu_{l}>0$ given in (Equation 31), it holds that
(48)$$\begin{array}{c} \left(\xi +1\right)\left(\sqrt{\frac{1-p_{l}}{1-p_{*}-\varepsilon }}\right)\frac{1}{\nu_{l}}=1. \end{array}$$
Hence, from (47) and (Equation 48), we have
(49)$$\begin{array}{c} L_{l}\left(W\right)1_{S_{l}}\geq \theta 1_{S_{l}}=\frac{2\delta }{1-4\gamma_{0}}1_{S_{l}}, \end{array}$$
for all l∈T(p,ε), by the definition of θ in (14).We are now ready to complete the proof:
(50)Pl[Blc]≥Pl[Sl](51)=El[1Sl](52)=E1Ll(W)1Sl(53)≥E12δ1−4γ01Sl(54)=2δ1−4γ0P1[Sl](55)>2δ1−4γ01−4γ02(56)=δ,
where (Equation 50) follows from the definition of set $S_{l}$ in (34), (52) follows by a standard change of measure [20], (53) follows from (Equation 49), and (55) follows from (Equation 36) of Lemma 1 (recall that we assumed (Equation 43)).The inequality (56) shows a contradiction to the fact that under $H_{l}$, the (ε,δ)-correct bandit policy must return the arm *l* with probability at least 1−δ, i.e., $P_{l}\left(B_{l}^{c}\right)\leq \delta$. This concludes the proof. □

From Lemma 2 and the definition of $G_{1,l}$ in (Equation 29), it holds that
(57)$$\begin{array}{c} E_{1}\left[\frac{7\alpha_{l}^{2}p_{l}^{2}{\left(1-p_{l}\right)}^{2}}{2\left(p_{*}+\varepsilon \right)\sqrt{\left(1-p_{l}\right)\left(1-\left(p_{*}+\varepsilon \right)\right)}}T_{l}\right]\geq \frac{\gamma_{0}}{\nu_{l}}\log\frac{1}{\theta } \end{array}$$
for all l∈T(p,ε). Hence, and using the definition of $\nu_{l}$ in (Equation 31), we have
(58)E1αl2pl2(1−pl)2Tl≥2γ0(p*+ε)(1−pl)(1−p*−ε)7(1+ξ)1−p*−ε1−pllog1θ(59)=2γ0(p*+ε)(1−p*−ε)7(1+ξ)log1θ.


*Step 3: Deducing a Lower Bound on the Sample Complexity*


For any arm l∈T(p,ε), by the definition of $\alpha_{l}$ in (Equation 25), we have
(60)$$\begin{array}{c} \alpha_{l}^{2}p_{l}^{2}{\left(1-p_{l}\right)}^{2}={\left(\varepsilon +\Delta_{l}\right)}^{2}. \end{array}$$
Note that $0\leq \Delta_{l}<\varepsilon$ for all $l\in M_{0}\left(p,\varepsilon \right)$, since $M_{0}\subseteq M$, the set of ϵ-optimal arms. Therefore, we can further simplify (Equation 60) to
(61)$$\begin{array}{c} \alpha_{l}^{2}p_{l}^{2}{\left(1-p_{l}\right)}^{2}\leq 4\epsilon^{2} \end{array}$$
for $l\in M_{0}\left(p,\varepsilon \right)$.

Substituting (Equation 60)–(Equation 61) into (59), we obtain
(62)Ep[T]=Ep∑l=1MTl(63)≥Ep∑l∈M0(p,ε)Tl+Ep∑l∈N(p,ε)Tl(64)≥2γ0(p*+ε)(1−p*−ε)7(ξ+1)|M0(p,ε)|4ε2+∑l∈N(p,ε)1(ε+Δl)2log1θ(65)=2γ0(p*+ε)(1−p*−ε)7(ξ+1)|M0(p,ε)|4ε2+∑l∈N(p,ε)1(ε+Δl)2log1+4δ04δ,
where (65) uses the definition of θ in (14). Finally, we obtain (Equation 12) from (65) and (Equation 19).

## 4. Conclusion

We have presented a refined analysis of best-arm identification following the gap-based approach of [14], but incorporating refinements that circumvent some weaknesses, leading to a bound matching the divergence-based approach [15] in many cases. It would be of interest to determine whether further refinements could allow this approach to match [15] in all cases, or the extent to which the gap-based approach extends beyond Bernoulli rewards and/or beyond the standard best-arm identification problem (e.g., to ranking problems [21]).

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
