# Peer review of "On Gap-Based Lower Bounding Techniques for Best-Arm Identification"

_entropy, 2020, doi:10.3390/e22070788_

Round 1
Reviewer 1 Report
This is a nice contribution clarifying the connection between two existing lower bounds for the best-arm identification problem and refining a lower bound by Mannor and Tsitsiklis. It merits publication. I have only minor comments:
- p. 2, line 52: what does 'PAC' stand for?
- p. 2, line 55: better define underline p first as the identifying parameter of the class of bandit instances that you consider. As written this parameter appears out of nowhere and is a bit confusing.
- In (6), is delta' equal to delta?
- (6), (9) and (12) look a bit confusing because as I understand the problem setting, these bounds should apply to any bandit instance, yet the quantity M(p, eps) defined in (3) is instance-specific. Do you mean to say that (6), (9) and (12) hold for any algorithm that is instance-agnostic, yet the bounds themselves depend on the problem instance? It might be worth clarifying this point.
- p. 3, line 58: better clarify above (6) what else c_1 depends on. The way it's written above (6) one gets an impression it is simply a constant that depends on underline p only.
- since in (12) there is no dependence on underline p, can comment on why the assumption 0 < underline p is still required?
- p. 4, line 70: what do you mean by 'tight' in 'the right-hand inequality in (10) is tight'?
Author Response
Please see the attached response document.

Reviewer 2 Report
Dear Author,
The work that you present is very interesting from a theoretical point of view (specifically for Multi-armed bandits and PAC Learning). Also, the theoretical framework is well contextualized. The article shows a difficult sequence to follow, but I think they are correct and original and should be published. I would like to suggest that you should emphasize: how does your work contribute to the applications that you mention? something more practical. Finally, you propose in your conclusions that "It would be of interest to determine whether further refinements could allow this approach to match [15] in all cases" Why not in this work? I think it would not be so difficult for them to do this.
Author Response

(The authors gave the same response as above.)
